# Duration of External Neck Stabilisation (DENS) following odontoid fracture in older or frail adults: protocol for a randomised controlled trial of collar versus no collar

Julie Woodfield ![ORCID],[1,2] Ellie Edlmann,[3,4] Polly L Black,[5] Julia Boyd,[6] Phillip Correia Copley ![ORCID],[1] Gina Cranswick,[6] Helen Eborall ![ORCID],[7] Catriona Keerie,[6] Sadaquate Khan,[1] Julia Lawton,[7] David J Lowe ![ORCID],[8,9] John Norrie ![ORCID],[6] Angela Niven,[6] Matthew J Reed,[5,7] Susan Deborah Shenkin ![ORCID],[7,10] Patrick Statham,[1] Andrew Stoddart,[11] James Tomlinson,[12] Paul M Brennan[1,2]

**Correspondence to**
Dr Paul M Brennan;
paul.brennan@ed.ac.uk

## ABSTRACT

**Introduction**  Fractures of the odontoid process frequently result from low impact falls in frail or older adults. These are increasing in incidence and importance as the population ages. In the UK, odontoid fractures in older adults are usually managed in hard collars to immobilise the fracture and promote bony healing. However, bony healing does not always occur in older adults, and bony healing is not associated with quality of life, functional, or pain outcomes. Further, hard collars can cause complications such as skin pressure ulcers, swallowing difficulties and difficulties with personal care. We hypothesise that management with no immobilisation may be superior to management in a hard collar for older or frail adults with odontoid fractures.

**Methods and analyses**  This is the protocol for the Duration of External Neck Stabilisation (DENS) trial—a non-blinded randomised controlled trial comparing management in a hard collar with management without a collar for older (≥65 years) or frail (Rockwood Clinical Frailty Scale ≥5) adults with a new odontoid fracture. 887 neurologically intact participants with any odontoid process fracture type will be randomised to continuing with a hard collar (standard care) or removal of the collar (intervention). The primary outcome is quality of life measured using the EQ-5D-5L at 12 weeks. Secondary outcomes include pain scores, neck disability index, health and social care use and costs, and mortality.

**Ethics and dissemination**  Informed consent for participation will be sought from those able to provide it. We will also include those who lack capacity to ensure representativeness of frail and acutely unwell older adults. Results will be disseminated via scientific publication, lay summary, and visual abstract. The DENS trial received a favourable ethical opinion from the Scotland A Research Ethics Committee (21/SS/0036) and the Leeds West Research Ethics Committee (21/YH/0141).

**Trial registration number**  NCT04895644.

## STRENGTHS AND LIMITATIONS OF THIS STUDY

⇒ The Duration of External Neck Stabilisation (DENS) trial is a multicentre non-blinded randomised controlled trial comparing the use of hard collars with no immobilisation.
⇒ The primary outcome measure (EQ-5D-5L) is a quality of life measure relevant to the older or frail population.
⇒ Health economic analysis and embedded qualitative study methods assess the wider implications of collar use in the older or frail population.
⇒ Inclusion of all odontoid process fracture types, frail patients and those with cognitive impairment will ensure the results are applicable to these populations.
⇒ The DENS trial does not directly compare surgical fixation and no immobilisation due to the preference for non-surgical management for older or frail patients in the UK.

## INTRODUCTION
### Background and rationale

Odontoid process (dens) fractures occur following low impact falls in frail and older people, and are increasing in incidence as the population ages.[1–3] The Trauma Audit and Research Network database identified at least 1700 odontoid fractures each year in England and Wales, 85% in people over 65 years. One-year mortality following an odontoid fracture is 20%–50%, similar to or higher than following a hip fracture, likely reflecting the underlying frailty and health status of those at risk of low impact falls.[4–7]

In the UK, 85%–90% of frail or older people with a new odontoid fracture are managed non-operatively, most with 6–12

weeks immobilisation in a hard collar.[8–10] This aims to promote bony fusion and prevent neurological deterioration or instability pain.[1 5 11] However, hard collars only restrict 40%–50% of neck movements,[12 13] and bony fusion rates of 20%–80% are variable across fracture types, management, age and frailty.[1 5 11 14–18] Further, bony fusion may not be associated with pain, quality of life (QoL), mortality or functional outcomes in older people,[8 19 20] and late neurological deterioration is very rare.[5 11 21] Hard collars can cause skin pressure ulcers and difficulties with swallowing, breathing and personal care, all of which can affect QoL.[10 22] Additional health and social care input is often required to assist patients with activities of daily living. If fibrous union, or lack of bony fusion is an acceptable outcome and hard collars can negatively impact QoL, then management of an odontoid fracture with a hard collar may be causing unnecessary harm and negatively impacting on QoL in a group for whom the short to medium term QoL may be more important than long term outcomes.

Spinal surgeons responding to a UK survey variably reported managing odontoid fractures without any immobilisation or removing hard collars in frail or older patients who were unable to tolerate a collar, had a short life expectancy or suffered complications.[10] Ninety percent supported randomisation to management with or without a collar to determine optimum treatment.[10] Further, feedback from patients managed in collars supports exploring whether collars are necessary for odontoid process fractures.

The Duration of External Neck Stabilisation (DENS) trial will test the hypothesis that management of odontoid process fractures without immobilisation is associated with improved QoL compared with management in a hard collar for 12 weeks.

## Aims and objectives

The primary aim is to determine the difference in QoL measured using the EQ-5D-5L (EuroQol 5 dimension instrument with 5 levels) at 12 weeks between frail or older patients with odontoid process fractures managed with a hard collar (standard care) or without any immobilisation (intervention).

Secondary aims include assessing differences in neck pain, functional recovery, complications, radiological findings, mortality and overall healthcare use and costs, up to 1 year postinjury.

A nested qualitative study will explore trial recruitment experiences from the perspectives of patients, carers, and healthcare professionals, and explore adherence to group allocation. Findings from the qualitative study will inform recruitment and consent procedures and aid interpretation of trial results.

## METHODS AND ANALYSIS
### Study design

This manuscript has been prepared in accordance with the SPIRIT checklist. The DENS trial is a non-blinded randomised controlled trial with a nested qualitative study comparing early hard collar removal (intervention) with treatment in a hard collar for 12 weeks (standard care) in older or frail adults with a new odontoid process fracture. Figure 1 shows the trial flow chart. The study will recruit from November 2021 to December 2023.

### Study eligibility
#### Inclusion criteria

► Aged ≥65 years, or Rockwood Clinical Frailty Scale (CFS) of ≥5 (at least mildly frail: help needed in high order instrumental activities of daily living).[23]
► Recent odontoid process fracture of any type[24] secondary to low impact trauma (any degree of fracture angulation, displacement or canal narrowing).
► History of recent trauma (within 3 weeks).
► Recruited within 3 weeks of injury.
► Determined by consultant spinal surgeon as suitable for standard treatment in hard collar or treatment without a collar.

#### Exclusion criteria

► Fracture sustained in a high-impact injury.
► New neurological deficit attributable to the fracture.
► Unable to tolerate a hard collar.
► Additional (non-odontoid process) cervical spine fracture not suitable for management without a hard collar.
► Underlying condition with risk of spinal instability (eg, ankylosing spondylitis).
► Fracture suspected to be older than 3 weeks at the time of assessment.
► Not expected to survive to hospital discharge.

A consultant radiologist report of an odontoid process fracture on CT and confirmation of eligibility by a consultant spinal surgeon (orthopaedic or neurosurgical) is required. Patients with additional cervical spine fractures are eligible provided the consultant spinal surgeon determines the fracture(s) are suitable for management without a hard collar. There are no exclusions for other injuries (eg, fractured femur). Co-enrolment in other studies is permitted where this does not affect the DENS trial randomisation allocation or outcome measure assessment, and where this is not expected to burden the participant.

### Participant recruitment

Potential participants will be identified by clinicians following presentation to emergency departments (EDs) or acute assessment units, through review of acute cervical spine CT reports, acute hospital attendance or through referrals or admissions to spinal services in the UK. An anonymised screening log will be kept.

Any neck stabilisation can be applied as part of standard care prior to randomisation, including; trauma collar, blocks, spinal board, padded hard collar (eg, Miami J, Aspen, Philadelphia) or soft collar. Standard care treatment, such as removal from spinal boards

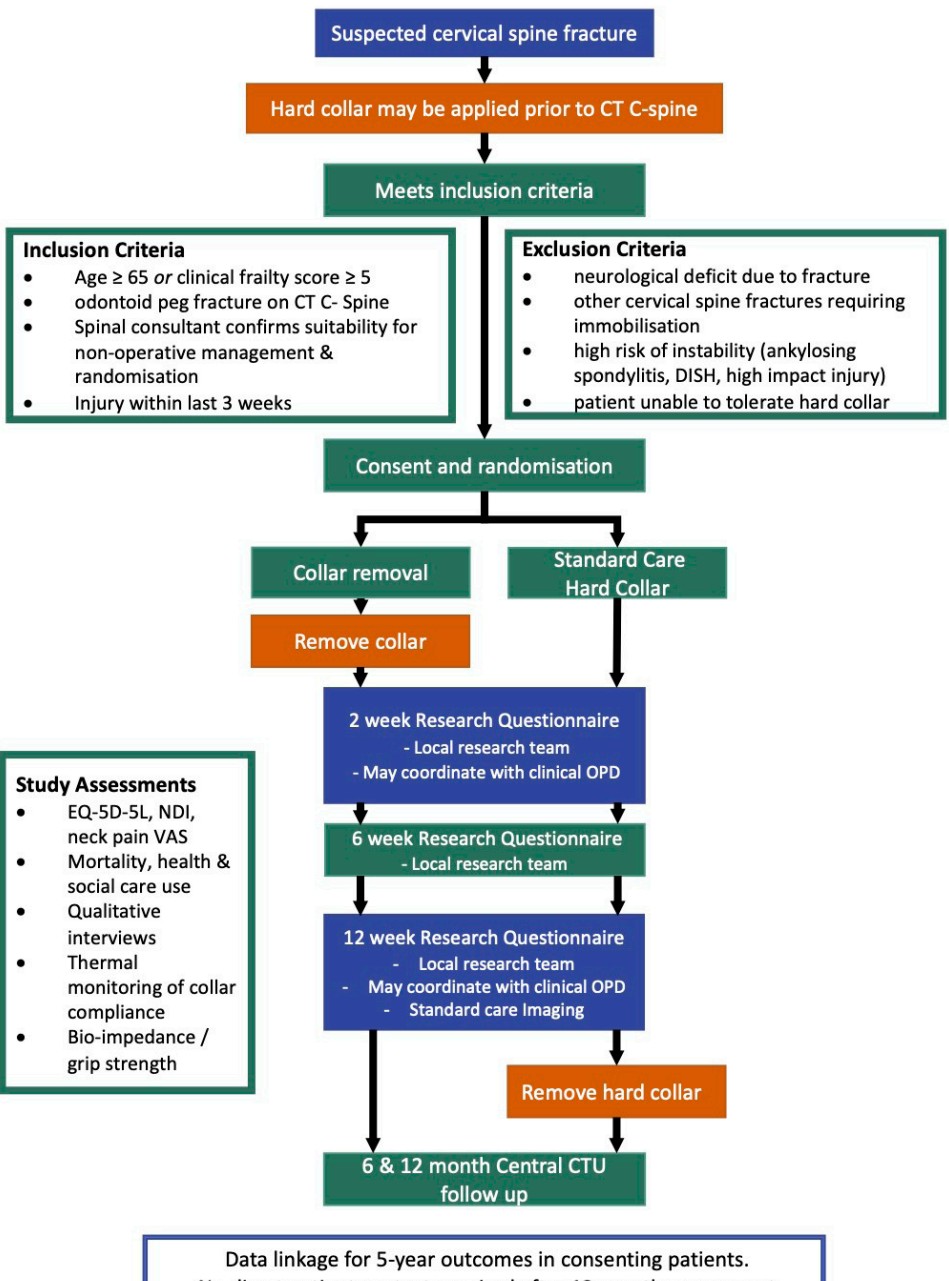

**DENS RCT**
**Duration of External Neck Stabilisation following odontoid fracture in older or frail adults:**
**a randomised controlled trial of collar versus no collar**

Suspected cervical spine fracture

Hard collar may be applied prior to CT C-spine

Meets inclusion criteria

**Inclusion Criteria**
- Age ≥ 65 *or* clinical frailty score ≥ 5
- odontoid peg fracture on CT C- Spine
- Spinal consultant confirms suitability for non-operative management & randomisation
- Injury within last 3 weeks

**Exclusion Criteria**
- neurological deficit due to fracture
- other cervical spine fractures requiring immobilisation
- high risk of instability (ankylosing spondylitis, DISH, high impact injury)
- patient unable to tolerate hard collar

Consent and randomisation

Collar removal

Standard Care Hard Collar

Remove collar

2 week Research Questionnaire
- Local research team
- May coordinate with clinical OPD

**Study Assessments**
- EQ-5D-5L, NDI, neck pain VAS
- Mortality, health & social care use
- Qualitative interviews
- Thermal monitoring of collar compliance
- Bio-impedance / grip strength

6 week Research Questionnaire
- Local research team

12 week Research Questionnaire
- Local research team
- May coordinate with clinical OPD
- Standard care Imaging

Remove hard collar

6 & 12 month Central CTU follow up

Data linkage for 5-year outcomes in consenting patients.
No direct patient contact required after 12-month assessment

**Figure 1** DENS trial flow chart. C-spine, cervical spine; CTU, Clinical Trials Unit; DENS, Duration of External Neck Stabilisation; DISH, diffuse idiopathic skeletal hyperostosis; NDI, Neck Disability Index; RCT, randomised controlled trial; VAS, Visual Analogue Scale; OPD, Outpatient Department; EQ-5D-5L, EuroQol 5 dimension instrument 5 level version.

and placement of a padded hard collar will occur as usual, and will not be delayed by trial assessment and participation.

Eligibility assessment and randomisation will take place as soon as possible after injury (target within 48 hours) to maximise the impact of study intervention. However, falls in older and frail people are usually multifactorial,[25 26] and conditions leading to the fall or resulting from the fall may affect capacity and fitness to participate. Recruitment can occur up to 3 weeks after the injury to provide flexibility for those presenting late, facilitate participation in those who are acutely unwell, and ensure time for involvement of patient representatives where needed.

The nested qualitative study will include healthcare professionals involved in patient recruitment, study participants, carers of participants lacking capacity and those who decline trial participation.

## Randomisation and interventions

Web-based randomisation will occur with a minimisation algorithm based on:

► Age at randomisation (<75 vs ≥75 years).
► Odontoid fracture type (I–III).[24]
► Frailty (CFS ≥5 vs <5).[23]

The first participant will be allocated to one arm with a probability of 0.5 and subsequent participants will be allocated with a probability of 0.8 to the group which minimises differences of the above variables between the two trial arms. Treatment allocation will be disclosed to the local research team via the web interface following randomisation.

The standard care arm will involve 12 weeks of hard collar treatment, following local usual choice of hard collar and local usual arrangements for collar care. For those allocated to the intervention (collar removal) arm, the collar will be removed as soon as possible, but may be weaned over several days if required for optimising analgesia. Removal of a collar prior to 12 weeks in the hard collar arm or wearing of a collar in the no collar arm will be recorded as a failure of adherence with explanation. Soft collars will not be used in either arm.

## Outcome measures

The primary outcome measure is the EQ-5D-5L at 12 weeks postrandomisation.

Secondary outcome measures are:

► Two and 6 weeks: EQ-5D-5L, Neck Disability Index (NDI), Neck Numeric Pain Scale.
► Twelve weeks: NDI, Neck Numeric Pain Scale, adverse events (AEs), compliance with hard collar use, analgesia use, length of hospital stay, discharge destination, bony fusion/stability on imaging, loss of muscle bulk in upper limbs using grip strength and bioimpedence.
► Six and 12 months: EQ-5D-5L, NDI, Neck Numeric Pain Scale, injury-related complications, mortality, health and social care visits/use, hospital admission or outpatient visits, health economics analysis.

**Table 1** Timing of study assessments

| | Timing from randomisation | | | | | | |
|---|---|---|---|---|---|---|---|
| | Baseline | Discharge | 2 weeks | 6 weeks | 12 weeks | 6 months | 12 months |
| Assessment | | | | | | | |
| Eligibility assessment | * | | | | | | |
| Consent | * | | | | | | |
| Demographics | | | | | | | |
| Baseline data | * | | | | | | |
| CFS[23] | * | | | | | | |
| Imaging | | | | | | | |
| Fracture type | X | | | | | | |
| Imaging type | X | | | | X | | |
| Outcomes | | | | | | | |
| EQ-5D-5L | | | * | * | * | * | * |
| EQ-VAS | | | * | * | * | * | * |
| Neck Numeric Pain Scale | | | * | * | * | * | * |
| NDI | | | * | * | * | * | * |
| AEs | | * | | | * | | |
| Injury-related complications | | | | | | * | * |
| Collar use | | * | * | * | * | | |
| Grip strength/bioimpedance | * | | | | * | | |
| Service use | | | * | * | * | * | * |
| Analgesia use | | | * | * | * | | |
| Mortality | | | | | * | * | * |

Assessments undertaken as standard care are shown with x. Study specific assessments are shown with*. Assessments may occur within the following time windows: 2 weeks (±1 week), 6 and 12 weeks (±2 weeks) and 6 and 12 months (±4 weeks). Fracture type refers to that of Anderson and D'Alonzo.[24]
EQ-5D-5L - EuroQol 5 dimension instrument 5 level version
AEs, adverse events; CFS, Clinical Frailty Scale; NDI, Neck Disability Index; VAS, Visual Analogue Scale.

## Trial assessments

Assessments and timing of data collection are shown in table 1. Baseline data on demographics, comorbidities, injury, radiological findings and inpatient management will be recorded. Patient diaries will be used to collect pain scores, analgesia and collar use. Standard local clinical follow-up will occur. Research assessments will be carried out at 2, 6 and 12 weeks by the local research team and by a blinded member of the central research team at 6 and 12 months. Assessments will occur via telephone interviews, postal or online questionnaires, or during standard care appointments. A proxy can provide information on a patient's behalf when required. Up to three reminder telephone calls will be made to maximise completion.

At centres with facilities for routine assessment of grip strength and bioimpedence, all participants will be invited to take part in this arm of the study. Loss of muscle bulk in the upper limbs will be assessed over 12 weeks using hand grip strength measured with a dynamometer.[27] Bioimpedence measurements will be made at the wrist and ankle. Muscle mass and skeletal muscle index as a marker of frailty will be calculated as per Janssen et al.[28]

Approximately 25 patients in the collar arm will use temperature sensors to assess compliance with the collar (Thermochron iButtons).[29] These will be the first participants willing to participate at centres where the temperature sensors can be missioned. Long-term follow-up of mortality and hospital admissions using anonymised data without participant contact will occur for 5 years.

No study specific follow-up imaging is mandatory. Participants will follow local clinical protocols for imaging. The modality and findings of all imaging undertaken will be recorded and imaging collected at the study centre for analysis where patient consent for transfer is given.

## Qualitative study

Qualitative researchers will interview healthcare professionals about their recruitment experiences, perceptions of why patients decline or consent to participation, and reasons for not approaching eligible patients. Patients and caregivers will be interviewed at two time points—following randomisation and 12 weeks later. Postrandomisation interviews will explore reasons for taking part or declining participation, views about trial recruitment approaches, hopes and expectations regarding trial participation, and any anticipated difficulties or concerns adhering to treatment allocation. Follow-up interviews will explore treatment adherence, benefits or burdens of the allocated treatment and perceptions of the impact on QoL and recovery. All interviews will use topic guides to help ensure the discussion remains relevant to addressing the study aims, while offering participants flexibility to raise issues they consider important, including those unforeseen at the study outset.

## Sample size

The recruitment target is 887 participants from around 25 UK sites over 2 years. An unadjusted sample size was

**Table 2** Study progression criteria

| | Category | | |
|---|---|---|---|
| | **Red** | **Amber** | **Green** |
| Total no recruited | ≤108 | 109–131 | ≥132 |
| Recruitment threshold (%) | ≤82 | 83–99 | 100 |
| Recruitment rate (patients per site per month) | 1.07 | 1.33 | 1.63 |
| No of sites open | <5 | 5–10 | >10 |
| Adherence failure | >20% | 11%–20% | ≤10% |

Study progression criteria for the internal pilot to assess recruitment feasibility after the first 9 months of recruitment. The traffic light categories will lead to the following actions. Green: continue unchanged. Amber: make changes, including adding more sites or increasing the time period for recruitment. Red: continue stopping as study may not be feasible, unless identifiable and rectifiable causes are identified. Adherence failure includes study withdrawals and cross-over (removing a collar in the collar arm, or wearing a collar in the no collar arm).

calculated based on 90% power and a 5% significance level, using a two-sided two-sample t-test, for an effect size of 1/6 (0.05 minimum clinically important difference/0.3 SD) on the EQ-5D-5L at 12 weeks. This results in a sample size of 1514, increasing to 1893 assuming 20% missing data. This sample size can be reduced by accounting for correlation between baseline covariates (age, CFS) and 12 week EQ-5D-5L. It is anticipated that this correlation will be around 0.5, allowing a 25% reduction in sample size. Higher correlation will allow a larger reduction.[30] Further, the postrandomisation measures of EQ-5D-5L at 2, 6 and 12 weeks, are assumed to have a serial correlation of 0.44 which allows a further 37.5% reduction in sample size.[30 31] Including these reductions leads to the final target sample size of 887, with 1:1 allocation.

To check these assumptions, a sample size re-estimation will be modelled using a simulation approach based on 12 week outcome data from the first 300 participants.[30] Due to the conservative assumptions made, particularly the assumed SD, the probability of increasing the sample size is small, and there is a larger probability it can be reduced.

Based on audits in two UK neurosurgical units, and assuming a 50% recruitment rate with staggered site opening, the target recruitment rate is 1.6 patients per site per month. An internal pilot will assess recruitment feasibility, targeting 132 patients (15% total) across 18 centres in 9 months. Progression criteria for the pilot are shown in table 2.

The qualitative study will recruit approximately 20 healthcare professionals and 30 patients or carers from a cross section of sites. Purposive sampling will be used to ensure representation of different ages, socioeconomic backgrounds, cognitions and residential setting among patients. Sampling decisions may be revised in light of emerging findings.

### Data analysis plan

The primary data analysis will be by intention to treat. A 5% two-sided significance level will be used throughout. A detailed statistical analysis plan will be developed by the study statisticians and finalised prior to locking of the trial database.

Analysis of the primary outcome will be a repeated measures analysis of covariance, including terms for treatment arm and the EQ-5D-5L responses at 2, 6 and 12 weeks. Adjustments will be made for the variables included in the randomisation minimisation algorithm. Adjustment for study site will be included as a random effect, if appropriate. The repeated measures approach enables estimation of an intervention effect at week 12 postrandomisation (primary outcome), and an overall assessment of the effect of the intervention over the 12-week period.

The 12-week NDI and Visual Analogue Scale will be analysed in a similar manner to the EQ-5D-5L. It is anticipated that results will underpin any between group differences seen in EQ-5D-5L, confirming and strengthening the clinical interpretation of the findings. All other secondary outcomes will be analysed with statistical models appropriate to the distribution of the outcome (see table 3). Where there are data recorded on multiple occasions postrandomisation, we may use an appropriate repeated measures model. Exploratory subgroup analyses by fracture type (I, II and III),[24] CFS (<5 vs ≥5)[23] and age (<75 vs ≥75 years) will be performed. The influence of any missing data on the robustness of the findings will

**Table 3** Possible methods of analysis

| Variable | Hypothesis | Outcome measures | Analysis methods |
|---|---|---|---|
| Baseline data | No difference between groups | Gender, age, comorbidities, injury data, radiological findings, inpatient management, CFS | Categorical variables: absolute numbers and percentages. Continuous variables: mean, median, SD, IQR. |
| **Primary outcome** | | | |
| Health-related quality of life at 12 weeks | Clinically important difference between two groups with early collar removal superior | EQ-5D-5L (continuous) | Repeated measures analysis of covariance at 2, 6 and 12 weeks, adjusting for minimisation variables |
| **Secondary outcomes** | | | |
| Pain | No difference between groups | NPRS (ordinal) | Ordinal regression |
| Function | No difference between groups | EQ-VAS (continuous), NDI (continuous) Grip strength (continuous) Bio impedance (ordinal) | Repeated measures analysis of covariance at 2, 6 and 12 weeks, adjusting for minimisation variables Student t-test Nonparametric methods; Mann-Whitney, Wilcoxon signed ranks test |
| Mortality | No difference between groups | All-cause mortality (binary) | Logistic regression, $X^2$ |
| Collar use | Significantly greater in the collar group | No of days of collar use (continuous) | Poisson regression |
| Health economics | Baysian assessment of likelihood incremental cost per QALY of either arm is below NICE Thresholds (£20 k to £30 k per QALY) | Service use & cost, QALYs (adjusted from EQ-5D-5L) Incremental Cost Per QALY | Generalised linear lodelling of cost and QALYs. Probabilistic sensitivity analysis via recycled predictions |
| **Sub-group analysis;** | | | |
| Fracture type (l, II and II) | No difference | | For each analysis, the subgroup variable and interaction term between intervention and subgroup variable will be included in the model. |
| CFS (<5 vs ≥5) | Better treatment effect in CFS ≥5 | | |
| Age (<75 vs ≥75 years) | Better treatment effect in age ≥75 | | |

Methods to be used for analysing variables collected in the DENS study. A detailed statistical analysis plan will be published.
CFS, Clinical Frailty Scale; DENS, Duration of External Neck Stabilisation; NDI, neck disability index; NICE, National Institute for Health and Care Excellence; NPRS, Numeric Pain Rating Scale; QALY, quality-adjusted life year; VAS, Visual Analogue Scale.

be examined, for example, using multiple imputation models under a missing at random assumption. Screening logs will be analysed to assess generalisability of results.

Interview data will be analysed thematically using the method of constant comparison. To ensure rigour, several individuals will be involved in data analysis and coding. An interactive working group will be set up, comprising the qualitative research team, co-investigators, the trial manager, patient and public involvement (PPI) representatives, and at least one healthcare professional from each of the sites involved in the pilot phase. A 'what, so what, now what' approach[32] will be used to translate qualitative findings and other experiences shared by group members into tangible recommendations.

A 12-month within trial health economic analysis will be undertaken based on National Institute for Health and Care Excellence (NICE) reference case recommendations to maximise UK policy relevance.[33] This will include: adoption of a National Health Service (NHS) and Personal Social Services decision perspective; cost-utility approach for primary analysis (results presented in terms of incremental cost per quality-adjusted life year (QALY), derived from EQ-5D-5L data with an area under the curve approach, omitting baseline); discount rate of 3.5% for both costs and QALYs (where applicable); and use of probabilistic sensitivity analysis, to generate cost-effectiveness acceptability curves via the recycled predictions technique.[33 34] Choice of primary analysis cost per QALY threshold and EQ-5D-5L scoring algorithm will match NICE preferences at the time of data lock. Resource use will be combined with standard UK price weights to generate costs.[35 36] The latest financial year for which at least one study participant provides data and prices are available will be selected as base year. Univariate mean QALYs, resource use and costs will be presented for each trial arm alongside differences in means and associated 95% CIs. Multivariate analysis of both QALYs and total costs, will also be presented controlling for baseline costs where available and minimisation variables. Missing data will be imputed using appropriate techniques depending on degree of missingness, likely multiple imputation by chained equations.

## Patient and public involvement

Patients and the public have played an active role in developing the grant application and project protocol, are co-applicants on the grant and are represented on the trial steering committee. We asked older people with odontoid fractures and members of a PPI group for their opinion on EQ-5D-5L and NDI as possible primary outcome measures. Older patients with odontoid fractures managed to complete the EQ-5D-5L, while some struggled with the NDI, and described some of the NDI domains as irrelevant. PPI input identified frailty rather than age as more influential of daily function, leading to inclusion criteria including both. A lay group also reviewed and rewrote the lay summary of the trial and had input into its likely acceptability. At the end of the study, we will produce a plain language summary of the study results and a visual abstract, in conjunction with participants and our PPI partners.

## ETHICS AND DISSEMINATION

The DENS trial received a favourable ethical opinion from the Scotland A Research Ethics Committee (REC) on the 10th June 2021 (21/SS/0036) and the Leeds West REC (on behalf of England and Wales) on the 29 July 2021 (21/YH/0141).

### Participant capacity and consent

The study member obtaining consent will explain and clarify all relevant information, either face to face, or via telephone or video call, and give patients the time they need to decide whether to participate. Informed consent will be signed and dated by participants or their representative, and a study team member. Verbal consent from a participant or their representative, signed and witnessed by a study team member will also be valid. Participants may withdraw consent at any time without reason, without affecting their care.

Patients diagnosed with odontoid fractures may lack capacity. Recruitment of these patients is necessary to aid generalisability of the trial outcome and to allow equal access to any benefits of participation. All consent procedures will adhere to the Adults with Incapacity (Scotland) Act 2000, the 2005 Mental Capacity Act, the Mental Capacity Act (Northern Ireland) 2016 and their codes of practice. For patients in Scotland who lack capacity, the patient's nearest relative, welfare attorney or guardian can provide consent. For patients in England, Wales and Northern Ireland who lack capacity, a personal consultee (relative or close friend) can provide consent. Where possible this will be written and signed, otherwise clear documentation of the discussion and named persons involved will be accepted.

If a participant without capacity regains capacity, informed consent will be sought to continue in the trial. In Scotland, if a participant has given informed consent, but loses capacity, they will remain in the study. In England, Wales, and Northern Ireland, if a participant has already given informed consent, but loses capacity, they will remain in the study only if consent is then given by a consultee.

### Safety considerations

AEs will be recorded up to 12 weeks. Expected AEs relating to hard collar use or odontoid process fractures include skin pressure ulcers, dysphagia and recurrent falls. All AEs (expected or not) will be reviewed by the local principal investigator and reported to the data monitoring committee (DMC). Serious AEs (SAEs) will also be reported to the trial sponsor, who will notify the REC. Outcome data, AEs and SAEs, will be routinely reviewed by the DMC. The DMC is independent and oversees trial safety and data monitoring. If the DMC assesses that the risk-benefit balance is significantly changed by any new safety information, amendments to the trial will be made.

The trial funder (NIHR) and cosponsors (University of Edinburgh and NHS Lothian) did not influence trial design. The sponsor ensures that data collection, management and trial monitoring are conducted appropriately and has overall responsibility for the study. Neither the funder nor the sponsor will have ultimate authority over writing of the report or the decision to submit the report for publication.

The trial will be coordinated by the central trial team based at the Edinburgh Clinical Trials Unit (ECTU), overseen by a Trial Management Group consisting of key trial members. Regular reports will be made to the Trial Steering Committee.

The protocol has been designed by the chief investigator and collaborators. The University of Edinburgh has insurance for negligent harm caused by poor protocol design, and participating sites have their own insurance and NHS indemnity for clinical negligence and other negligent harm.

## Data management

Participant records will be identified using a unique study identification number. All records will be kept in a secure storage area with limited access. Data will be entered anonymously into a secure database created and maintained by ECTU, stored on a secure server at The University of Edinburgh compliant with all relevant regulations. Trained and delegated members of the research team will have password protected logins for their own centre. Data generated from the iButton sensors is anonymised using a numerical code, which will be recorded on the database. Interviews for the qualitative study will be digitally recorded using a device encrypted to AES256 standard. Imaging will be stored and transferred within the NHS network. Only anonymised scans will be processed outside the NHS network. All investigators and study site staff will comply with the requirements of the appropriate data protection legislation (including the General Data Protection Regulation and Data Protection Act 2018) regarding collection, storage, processing and disclosure of personal information. Published results will not contain any personal data and individuals will not be identifiable. All trial related data will be archived for 5 years.

## Study dissemination

Ownership of the data arising from this study resides with the study team. The primary trial publication will be drafted by a writing committee and will recognise the work of those involved in the trial. PIs and investigators on delegation logs for patient recruitment will be invited as collaborators on the primary trial publication. Trial results will be published in a peer-reviewed journal, presented at meetings and published as a plain language summary and visual abstract.

**Author affiliations**
[1]Department of Clinical Neurosciences, NHS Lothian, Edinburgh, UK
[2]Translational Neurosurgery, The University of Edinburgh Centre for Clinical Brain Sciences, Edinburgh, UK
[3]Southwest Neurosurgical Centre, Derriford Hospital, Plymouth, UK
[4]Peninsula Medical School, University of Plymouth, Plymouth, UK
[5]Emergency Medicine Research Group (EMERGE), NHS Lothian, Edinburgh, UK
[6]Edinburgh Clinical Trials Unit, University of Edinburgh, Edinburgh, UK
[7]University of Edinburgh Usher Institute of Population Health Sciences and Informatics, Edinburgh, UK
[8]Department of Emergency, Queen Elizabeth University Hospital Campus, Glasgow, UK
[9]University of Glasgow Institute of Health and Wellbeing, Glasgow, UK
[10]Advanced Care Research Centre, University of Edinburgh College of Medicine and Veterinary Medicine, Edinburgh, UK
[11]Edinburgh Health Services Research Unit, The University Of Edinburgh, Edinburgh, UK
[12]Sheffield Teaching Hospitals NHS Foundation Trust, Sheffield, UK

**Acknowledgements** We thank all patients, lay members, and members of patient and public involvement groups who gave feedback to improve the study and design a study that is relevant and helpful for older or frail patients with odontoid process fractures.

**Contributors** JW, EE, PCC, PS, PMB contributed to the conception of the study. JW, EE, PLB, JB, PCC, GC, HE, CK, SK, JL, DJL, JN, AN, MJR, SDS, PS, AS, JT, and PBM all designed the protocol. JW, EE, JB, PCC, HE, CK, JL, DJL, MJR, SDS and AS drafted the protocol. JW, EE, PLB, JB, PCC, GC, HE, CK, SK, JL, DJL, JN, AN, MJR, SDS, PS, AS, JT, and PMB revised the manuscript critically for intellectual content and approved the final version for publication. PMB agrees to be accountable for all aspects of the work in ensuring that questions related to the accuracy or integrity of any part of the work are appropriately investigated and resolved.

**Funding** This work was supported by the National Institute for Health Research (NIHR) Health Technology Assessment (HTA) grant number 131118. EE, JW, MJR, CK, JN, DJL, SDS, and PFXS are supported by this grant.

**Disclaimer** The funder had no role in the study design, writing of the protocol, or the decision to submit the paper for publication.

**Competing interests** MJR is supported by an NHS Research Scotland Career Researcher Clinician award. EE is supported by Plymouth Charitable Fund and a University of Plymouth pump priming grant. All other authors declared no competing interests.

**Patient and public involvement** Patients and/or the public were involved in the design, or conduct, or reporting, or dissemination plans of this research. Refer to the Methods section for further details.

**Patient consent for publication** Not applicable.

**Provenance and peer review** Not commissioned; externally peer reviewed.

**ORCID iDs**
Julie Woodfield http://orcid.org/0000-0003-3645-500X
Phillip Correia Copley http://orcid.org/0000-0003-0706-6902
Helen Eborall http://orcid.org/0000-0002-6023-3661
David J Lowe http://orcid.org/0000-0003-4866-2049
John Norrie http://orcid.org/0000-0001-9823-9252
Susan Deborah Shenkin http://orcid.org/0000-0001-7375-4776

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
