## [Reviewer comments · BMJ Open]

ARTICLE DETAILS

TITLE (PROVISIONAL)	Duration of External Neck Stabilisation (DENS) following odontoid fracture in older or frail adults: protocol for a randomised controlled trial of collar versus no collar
AUTHORS	Woodfield, Julie; Edlmann, Ellie; Black, Polly; Boyd, Julia; Copley, Phillip; Cranswick, Gina; Eborall, Helen; Keerie, Catriona; Khan, Sadaquate; Lawton, Julia; Lowe, David; Norrie, John; Niven, Angela; Reed, Matthew; Shenkin, Susan; Statham, Patrick; Stoddart, Andrew; Tomlinson, James; Brennan, Paul

VERSION 1 – REVIEW

REVIEWER	Robinson, Anna-Lena Stockholm Spine Center
REVIEW RETURNED	06-Dec-2021

GENERAL COMMENTS	The authors must be commended that they target to study one of the major unanswered questions in cervical trauma. Unfortunately, the study design has some flaws and should not be published in its current form. Major issues are: • There is something fundamentally deviant with the research question as it insinuates that non-surgical treatment is generally accepted for individuals older than 65 years. Recent population-based studies found improved survival after odontoid fractures if treated surgically when younger than 88 years of age. This has not been addressed.• Surgeons based exclusion after enrolment, which can lead to a selection bias with more displaced type two fractures not included in the study. Detailed criticism: Title The title does not reflect the content of the protocol, the duration of the neck immobilisation is not compared in the study. Strengths & Limitations This section needs to be re-worked. There are no clear strengths or limitations mentioned. As this is a protocol article, the authors should mention whether or not they are following a protocol (i.e. SPIRIT), and which one. Inclusion criteria It is unclear whether point 4 overrides point 2, will this lead to a selection bias? (Spinal surgeon determines whether or not the patient is suitable for treatment in a collar or without a collar)
---

	The patients with a displaced fracture are more prone to become object for surgery or collar treatment, how will the authors decide which patients to include? Which criteria do the authors apply for surgical treatment indication? How will the authors adjust for differences in the distribution of type 2 and 3 fractures between the two treatment arms, if the authors do not differ, explain why. Participant recruitment If the authors recruit some of the patients after 3 weeks to the study, how will the authors treat the patients in the 3 first weeks? Collar/no collar? Some of the patients with a type 3 fracture will almost heal in 3 weeks and therefore the baseline data/ results will be inaccurate. The authors could consider enrolling the patients when admitting to the hospital. The authors should declare how to account for patients declining to participate. Are the authors going to list them and analyse i.e. what kind of fracture they have, and treatment they receive? Randomisation and Intervention The authors should explain why they chose their type of randomisation instead of traditional randomisation and support their reasoning with references. Outcome measures Please state what kind of follow up imaging will be used. CT or x-ray? Trial assessment Grip strength as measurement, please insert reference and how you will interpretate CFS based on grip strength. Define whom of the patients in the collar group will receive temperature sensors. Insert questionnaire for the qualitative study for the healthcare professionals etc. Statistics There is only little information on the planned statistics. Insert a table with including variables, hypothesis, outcome measures, method of analysis.
--	---

REVIEWER	Lofrese, Giorgio "M. Bufalini" Hospital
REVIEW RETURNED	16-Jan-2022

GENERAL COMMENTS	I express my personal congratulations to the authors for having conceived such a protocol for a randomized controlled trial in odontoid fractures. I would better specify the criteria regulating when removing collar in the "intervention" group, because I understand from the text that physicians could adapt their strategies from patient to patient, while from the table 1 it seems that they could choose to remove the collar at discharge, 2 weeks, or 6 weeks. Either in a way or in another the message should be clearly understandable, so I would improve the manuscript in this sense. I congratulate the authors even for having considered all the most relevant aspects determining a proper evaluation of functional outcome and quality
--

	of life, but I would extend the time for their “safety considerations”, because 12 weeks could be worthy if taking in account skin ulcers, dysphagia and falls, but we can’t ignore delayed neurological complications, which should be considered up to years after the traumatic event. Do the authors plan another study on that? On this field in which way and for how long the radiological outcome, correctly among the second aims of the trial, will be evaluated? The authors correctly mention the “stable non-union” as one of the principal concept at the base of their purpose, but how they plan to establish such a condition radiologically and at what time intervals? I’ve really appreciated the set up of your protocol, but I would improve it better specifying the aforementioned aspects.
--	---

VERSION 1 – AUTHOR RESPONSE

Reviewer: 1

Mrs. Anna-Lena Robinson, Stockholm Spine Center, Akademiska sjukhuset

Comments to the Author:

The authors must be commended that they target to study one of the major unanswered questions in cervical trauma. Unfortunately, the study design has some flaws and should not be published in its current form.

Major issues are:

- There is something fundamentally deviant with the research question as it insinuates that non-surgical treatment is generally accepted for individuals older than 65 years. Recent population-based studies found improved survival after odontoid fractures if treated surgically when younger than 88 years of age. This has not been addressed.

Thank you for your comments and review.

Our study includes only those where the responsible spinal consultant has already determined that surgical treatment is not appropriate and that conservative treatment without surgery is appropriate.

Surgery is rarely offered for those over 65 years with odontoid process fractures in the UK. Evidence of this can be seen in recent case series (see reference no 8, McIlroy et

al: <https://doi.org/10.1093/neuros/nyaa256>

) and also surveys of current practice (see reference no 9 Watts et

al: <https://doi.org/10.1016/j.injury.2021.09.057>). We understand that decision-making and

treatment differs in other countries where surgery may be more frequently employed and that the reviewer is leading a study in Sweden comparing surgical and non-surgical treatments.

(<https://trialsjournal.biomedcentral.com/articles/10.1186/s13063-018-2690-8>) Our aim is not to compare surgical and non-surgical treatment but to determine whether or not a hard collar is beneficial in those treated without surgery.

The question of whether or not to immobilise older people with odontoid process fractures in a hard collar was specifically determined in the UK to be an issue of importance for the population and therefore the theme of a National Institute for Health Research (NIHR) funding call. Our study design was selected in open competition for this funding.

- Surgeons based exclusion after enrolment, which can lead to a selection bias with more displaced type two fractures not included in the study.

The intention is to have a pragmatic study whereby patients who would normally be treated in a collar for 12 weeks are randomised to treatment with and without a collar. We specifically do not want to include patients who are considered for surgery – i.e. those at high risk without immobilisation or fixation. The exclusion happens at the time of determining eligibility for the study, not after enrolment. We will record displacement, angulation, and fracture type to assess the characteristics of patients that the treating teams usually manage conservatively and are willing to include in the study. The results of this study will only be applicable to those with characteristics similar to those included in the study. Those who would be managed with surgery will continue to be managed with surgery, and will not be eligible for inclusion in this study.

Detailed criticism:

Title

The title does not reflect the content of the protocol, the duration of the neck immobilisation is not compared in the stud.

In the UK, most adults with suspected cervical spine injuries are immobilised at the scene of the injury, or after presentation to the Emergency Department and before imaging is acquired. This is current practice for safety reasons prior to imaging being reviewed and spinal surgeons being consulted. Our study protocol refers to early removal of this initial immobilisation (within 3 weeks) versus prolonged immobilisation as a form of treatment for the fracture (for 12 weeks).

Strengths & Limitations

This section needs to be re-worked. There are no clear strengths or limitations mentioned. As this is a protocol article, the authors should mention whether or not they are following a protocol (i.e. SPIRIT), and which one.

We have adjusted the Strengths and Limitations section as per the instructions above in the Editor's comments. The SPIRIT checklist is included with the submission and we have referred to this in the Methods section.

Inclusion criteria

It is unclear whether point 4 overrides point 2, will this lead to a selection bias? (Spinal surgeon determines whether or not the patient is suitable for treatment in a collar or without a collar)

The patients with a displaced fracture are more prone to become object for surgery or collar treatment, how will the authors decide which patients to include? Which criteria do the authors apply for surgical treatment indication?

How will the authors adjust for differences in the distribution of type 2 and 3 fractures between the two treatment arms, if the authors do not differ, explain why.

We will only recruit patients who would not be treated with surgical intervention. We have deliberately not included particular Anderson and D'Alonzo subtypes or degrees of angulation or displacement in the inclusion criteria as surgeon practice varies and the aim is to reflect real world practice, and include those currently managed conservatively. If a patient would not be offered surgery based on their overall health, frailty, or fracture characteristics, they can be included in the study. Participating centres may have different thresholds for recruitment of patients into the study due to varying surgical strategies. We will record fracture type and clinical details. Exploratory subgroup analyses by fracture type (I, II and III), clinical frailty scale (<5 versus ≥5), and age (<75

versus ≥ 75 years) will be performed, as described in the methods section. Study site will be included as a random effect, if appropriate.

Participant recruitment

If the authors recruit some of the patients after 3 weeks to the study, how will the authors treat the patients in the 3 first weeks? Collar/no collar? Some of the patients with a type 3 fracture will almost heal in 3 weeks and therefore the baseline data/ results will be inaccurate. The authors could consider enrolling the patients when admitting to the hospital.

We will not recruit patients to the study more than 3 weeks following injury. We aim to recruit participants within 48 hours after injury and this is made clear at site initiation visits. However, we recognise that this is not always practically possible, particularly when participants may be medically unwell, in pain, frail, or have sustained other injuries. Participants may also present a few days following a low impact fall. We allow recruitment up to 3 weeks post injury to maximise recruitment whilst minimising the possibility of the initial treatment strategy influencing study results. In a survey of current UK practice carried out during design of the protocol, we did not identify any centres where fracture treatment/immobilisation was planned as being for less than six weeks duration following the injury.

Our methods section states:

Eligibility assessment and randomisation will take place as soon as possible after injury (target within 48 hours) to maximise the impact of study intervention. However, falls in older and frail people are usually multifactorial, and conditions leading to the fall or resulting from the fall may affect capacity and fitness to participate. Recruitment can occur up to 3 weeks after the injury to provide flexibility for those presenting late, facilitate participation in those who are acutely unwell, and ensure time for involvement of patient representatives where needed.

We anticipate that the majority of participants will be managed in a collar initially as this is standard trauma practice. Those who do not tolerate a collar or where a collar is deemed to be inappropriate (e.g palliative patients) will not be randomised to the study as not being able to tolerate a collar is an exclusion criteria. Thus, we will be comparing early collar removal (within 3 weeks) to late collar removal (at 12 weeks).

The authors should declare how to account for patients declining to participate. Are the authors going to list them and analyse i.e. what kind of fracture they have, and treatment they receive?

As stated in the methods section, we will keep an anonymised screening log. This will include demographic details, fracture characteristics, and reasons for ineligibility or participant refusal for participation. This data will be reported. We will also invite those who decline to participate in the trial to discuss their decision making with the qualitative research team to understand reasons for deciding not to take part. The qualitative study is designed to assess trial recruitment strategies and is described in the methods section.

Randomisation and Intervention

The authors should explain why they chose their type of randomisation instead of traditional randomisation and support their reasoning with references.

We use a standard randomisation technique with a minimisation algorithm. This is considered to be more robust than 1:1 allocation without minimisation or stratification, and is a standard method used commonly in randomised controlled trials. (see this BMJ Statistics Notes article: <https://www.ncbi.nlm.nih.gov/pmc/articles/PMC556084/>)

Outcome measures

Please state what kind of follow up imaging will be used. CT or x-ray?

No standard study specific imaging is required at centres. This is because there is currently no consensus on routine follow up imaging across the UK. Follow up imaging will be performed as per local arrangements for standard care. The aim of this study is not to assess imaging outcomes but to assess patient reported quality of life outcomes. Any imaging performed will be reported locally, recorded in the CRF, and also transferred to the study team for assessment.

We have added the following statement to make this clearer:

No study specific follow up imaging is mandatory. Participants will follow local clinical protocols for imaging. The modality and findings of all imaging undertaken will be recorded and imaging collected at the study centre for analysis where patient consent for transfer is given.

Trial assessment

Grip strength as measurement, please insert reference and how you will interpretate CFS based on grip strength.

Define whom of the patients in the collar group will receive temperature sensors.

Insert questionnaire for the qualitative study for the healthcare professionals etc.

We have inserted the following information into the protocol:

At centres with facilities for routine assessment of grip strength and bio-impedence, all participants will be invited to take part in this arm of the study. Loss of muscle bulk in the upper limbs will be assessed over 12 weeks using hand grip strength measured with a dynamometer. Bio-impedence measurements will be made at the wrist and ankle. Muscle mass and skeletal muscle index as a marker of frailty will be calculated as per Janseen et al. Approximately 25 patients in the collar arm will use temperature sensors to assess compliance with the collar (Thermochron iButtons). These will be the first participants willing to participate at centres where the temperature sensors can be missioned.

We do not intend to interpret the clinical frailty score based on grip strength, but to use grip strength as a proxy measure of frailty against which to interpret the outcomes of the study.

The qualitative study uses an open-ended interview design, not a questionnaire design. Hence, it is not possible for us to respond to this request to insert a questionnaire. We have revised the manuscript to make our qualitative study design clearer as below:

Qualitative researchers will interview healthcare professionals about their recruitment experiences, perceptions of why patients decline or consent to participation, and reasons for not approaching eligible patients. Patients and caregivers will be interviewed at two time points – following randomisation and 12 weeks later. Post-randomisation interviews will explore reasons for taking part or declining participation, views about trial recruitment approaches, hopes and expectations regarding trial participation, and any anticipated difficulties or concerns adhering to treatment allocation. Follow-up interviews will explore treatment adherence, benefits or burdens of the allocated treatment, and perceptions of the impact on QoL and recovery. All interviews will use topic guides to help ensure the discussion remains relevant to addressing the study aims, while offering participants flexibility to raise issues they consider important, including those unforeseen at the study outset.

Statistics

There is only little information on the planned statistics. Insert a table with including variables, hypothesis, outcome measures, method of analysis.

The statistical analysis of the primary outcome is detailed in the methods section as below:

Analysis of the primary outcome will be a repeated measures analysis of covariance, including terms for treatment arm and the EQ-5D-5L responses at 2, 6 and 12 weeks. Adjustments will be made for the variables included in the randomisation minimisation algorithm. Adjustment for study site will be included as a random effect, if appropriate. The repeated measures approach enables estimation of an intervention effect at week 12 post-randomisation (primary outcome), and an overall assessment of the effect of the intervention over the 12-week period.

We have not detailed all planned statistical analyses in detail due to the limited space and word count in the protocol article type. However, we plan to publish a detailed statistical analysis plan prior to data analysis as stated in the manuscript.

Reviewer: 2

Dr. Giorgio Lofrese, "M. Bufalini" Hospital

Comments to the Author:

I express my personal congratulations to the authors for having conceived such a protocol for a randomized controlled trial in odontoid fractures.

I would better specify the criteria regulating when removing collar in the "intervention" group, because I understand from the text that physicians could adapt their strategies from patient to patient, while from the table 1 it seems that they could choose to remove the collar at discharge, 2 weeks, or 6 weeks. Either in a way or in another the message should be clearly understandable, so I would improve the manuscript in this sense.

Thank you for your kind comments.

Participants will be randomised either to the intervention arm (removal of collar immediately) or the standard treatment arm (removal of collar at 12 weeks). It is not possible in this study to remove the collar at 6 weeks. The time points discharge, 2-weeks and 6-weeks only relate to time-points for collecting outcome data. We are encouraging centres to randomise patients, and remove collars in the intervention arm as soon as possible (within 48 hours), but allow up to 3 weeks to facilitate recruitment.

Our protocol states:

Eligibility assessment and randomisation will take place as soon as possible after injury (target within 48 hours) to maximise the impact of study intervention. However, falls in older and frail people are usually multifactorial, and conditions leading to the fall or resulting from the fall may affect capacity and fitness to participate. Recruitment can occur up to 3 weeks after the injury to provide flexibility for those presenting late, facilitate participation in those who are acutely unwell, and ensure time for involvement of patient representatives where needed.

I congratulate the authors even for having considered all the most relevant aspects determining a proper evaluation of functional outcome and quality of life, but I would extend the time for their "safety considerations", because 12 weeks could be worthy if taking in account skin ulcers, dysphagia and

falls, but we can't ignore delayed neurological complications, which should be considered up to years after the traumatic event. Do the authors plan another study on that?

We will assess patient outcomes and adverse events for one year post injury. We also seek consent at the time of participation for 5 year anonymous data linkage follow up so will be able to identify any serious longer term complications. Twelve weeks was selected for the primary end point as this is the end of treatment in the standard care arm, and reflects acute fracture and treatment morbidity rather than morbidity and mortality due to underlying health conditions, age, and frailty.

On this field in which way and for how long the radiological outcome, correctly among the second aims of the trial, will be evaluated? The authors correctly mention the "stable non-union" as one of the principal concept at the base of their purpose, but how they plan to establish such a condition radiologically and at what time intervals?

The radiological outcomes will be assessed through the case record form, with each participant undergoing usual local follow up. Types of imaging performed and imaging findings will be recorded. Imaging will also be transferred to the study team for assessment. No definitions or particular imaging modalities to assess fracture stability have been defined as local policies on standard imaging follow up vary widely across the UK for older or frail patients with odontoid process fractures. We have added the following statement:

No study specific follow up imaging is mandatory. Participants will follow local clinical protocols for imaging. The modality and findings of all imaging undertaken will be recorded and imaging collected at the study centre for analysis.

I've really appreciated the set up of your protocol, but I would improve it better specifying the aforementioned aspects.

Thank you very much for your time reviewing our protocol and for your comments. We hope we have satisfactorily addressed them.

Reviewer: 1

Competing interests of Reviewer: No conflicts of interest

We are aware of Reviewer 1's current study comparing surgical and non-surgical treatment for odontoid process fractures ((<https://trialsjournal.biomedcentral.com/articles/10.1186/s13063-018-2690-8>))

Reviewer: 2

Competing interests of Reviewer: Nothing to disclose

VERSION 2 – REVIEW

REVIEWER	Robinson, Anna-Lena Stockholm Spine Center
REVIEW RETURNED	20-Apr-2022

GENERAL COMMENTS	Thank you for the changes. I still have some minor things for you to consider. Strengths and limitations: Thank you for adding some strengths with the study. I still can't see any limitations mentioned. It will increase your credibility if you add some limitations along with the strengths. Statistics: It will be easier to overview and also to reproduce the study if you insert a table with the statistics, including variables, hypothesis, outcome measures, method of analysis.
--

REVIEWER	Lofrese, Giorgio "M. Bufalini" Hospital
REVIEW RETURNED	31-Mar-2022

GENERAL COMMENTS	The manuscript has been significantly improved reaching a level, which makes it acceptable for publication
--

VERSION 2 – AUTHOR RESPONSE

Response to reviewers

Reviewer: 2

Dr. Giorgio Lofrese, "M. Bufalini" Hospital

Comments to the Author:

The manuscript has been significantly improved reaching a level, which makes it acceptable for publication

Thank you for your comments and review.

Reviewer: 1

Mrs. Anna-Lena Robinson, Stockholm Spine Center, Akademiska sjukhuset

Comments to the Author:

Thank you for the changes. I still have some minor things for you to consider.

Strengths and limitations: Thank you for adding some strengths with the study. I still can't see any limitations mentioned. It will increase your credibility if you add some limitations along with the strengths.

We have added the following limitation to the strengths and limitations section:

- *The DENS trial does not directly compare surgical fixation and no immobilisation due to the preference for non-surgical management for older or frail patients in the UK.*

Statistics: It will be easier to overview and also to reproduce the study if you insert a table with the statistics, including variables, hypothesis, outcome measures, method of analysis.

We have added a table of outcome variables. This is based on the table in the reviewer's own protocol for a similar study (Robinson et al. Trials (2018) 19:452). We plan to publish a separate statistical analysis plan for the trial.

Reviewer: 2

Competing interests of Reviewer: Nothing to disclose

Reviewer: 1

Competing interests of Reviewer: none